# A handheld luminometer with sub-attomole limit of detection for distributed applications in global health

Paul Lebel[1]*, Susanna Elledge[2☯¤], Diane M. Wiener[1☯], Ilakkiyan Jeyakumar[1☯], Maíra Phelps[1], Axel Jacobsen[1], Emily Huynh[1], Chris Charlton[1], Robert Puccinelli[3], Prasenjit Mondal[4], Senjuti Saha[4], Cristina M. Tato[1], Rafael Gómez-Sjöberg[1]

**1** Chan Zuckerberg Biohub San Francisco, San Francisco, California, United States of America, **2** University of California, San Francisco, California, United States of America, **3** University of California, Berkeley, California, United States of America, **4** Child Health Research Foundation, Dhaka, Bangladesh

☯ These authors contributed equally to this work.
¤ Current address: Koch Institute, Massachusetts Institute of Technology, Cambridge, Massachusetts, United States of America
* paul.lebel@czbiohub.org

**Data Availability Statement:** All relevant data are within the paper and its Supporting Information files.

## Abstract

Luminescence is ubiquitous in biology research and medicine. Conceptually simple, the detection of luminescence nonetheless faces technical challenges because relevant signals can exhibit exceptionally low radiant power densities. Although low light detection is well-established in centralized laboratory settings, the cost, size, and environmental requirements of high-performance benchtop luminometers are not compatible with geographically-distributed global health studies or resource-constrained settings. Here we present the design and application of a ~$700 US handheld, battery-powered luminometer with performance on par with high-end benchtop instruments. By pairing robust and inexpensive Silicon Photomultiplier (SiPM) sensors with a low-profile shutter system, our design compensates for sensor non-idealities and thermal drift, achieving a limit of detection of 1.6E-19 moles of firefly luciferase. Using these devices, we performed two pilot cross-sectional serology studies to assess sars-cov-2 antibody levels: a cohort in the United States, as well as a field study in Bangladesh. Results from both studies were consistent with previous work and demonstrate the device's suitability for distributed applications in global health.

## Introduction

Highly sensitive, photomultiplier tube (PMT)—based commercial benchtop luminescence readers have been available for decades [1] and have enabled a myriad of luminescence-based assays for use in cell biology, hygiene applications, ATP sensing, and clinical applications [2]. These assays are easily employed in laboratories where infrastructure and skilled lab personnel can be centralized and where patients or study subjects can easily interact with their health

**Funding:** This work was funded by the Chan Zuckerberg Biohub. The funders had no role in study design, data collection and analysis, decision to publish, or preparation of the manuscript. PL, DW, IJ, MP, AJ, EH, CC, CT, and RGS earn salaries from Chan Zuckerberg Biohub.

**Competing interests:** S.E. declares a previously filed provisional patent application on the solution-based spLUC assay.

system [3]. However, access to these luminescence-based assays is significantly decreased in rural regions, areas populated by nomadic and marginalized groups, or where health infrastructure is non-existent [3,4] and incumbent instruments cannot meet the requirements of these regions. The need for point-of-care assays that can deliver rapid and actionable information that can be applied to management of endemic diseases and emerging infections is becoming increasingly apparent [4–7]. Portable, rugged equipment and simplified protocols for serological or bio-sensing assays are required to overcome these significant hurdles in order to provide rapid disease surveillance and management.

Engineered luminescent biosensors utilize a novel split-luciferase assay that leverages antibody specificity to recombine luciferase subunits to produce light [8]. This provides a highly-specific, quantitative readout of antibody levels in blood [8,9]. Being reconfigurable and simple to perform, the assay enables new opportunities for distributed, quantitative serology in areas where community healthcare workers and researchers travel to remote resource-limited areas across different geographical regions with little-to-no infrastructure. Ruggedized handheld luminescence readers that do not compromise on performance are therefore poised to become a broadly applicable tool for use in such communities. In addition to upholding the sensitivity of commercial instruments, the luminometers must meet several design requirements, including being: (1) portable and rugged; (2) able to operate outside of a temperature-controlled laboratory; (3) battery-powered; (4) low cost per unit for multiple instruments in simultaneous use; and (5) uniform in performance across devices.

Luminescence detection is conceptually simpler than fluorescence because it does not require excitation sources or spectral filtering. Nonetheless, there are unique engineering challenges related to the low radiant power density resulting from kinetically-limited enzymatic light generation. While a single fluorescent dye molecule is capable of emitting tens of thousands of photons per second from a nanoscopic volume [10,11], luminescence is generated gradually by enzymatic turnover on the order of a few photons per second per enzyme, even in optimized flash-mode assay conditions [12]. Consequently, a much larger number of emitting molecules is required to generate comparable photon rates by catalytic turnover as compared with fluorescence, placing a much greater demand on the design and sensitivity of luminometers relative to fluorescence readers. With large detector areas required to efficiently capture luminescence emission (S4 File), PMTs are the dominant technology, exhibiting large detector areas with exceptionally low dark current. A plurality of high-performing PMT-based commercial benchtop luminometers exist (S1 Table). Commercial instruments have long set a high standard for sensitivity–one example, the Promega GloMax Navigator system exhibits a limit of detection of 3E-21 moles of luciferase [13]. While commercial benchtop luminometers make demanding assays possible, the translation of these assays from the laboratory to a field setting is challenging, because commercial instruments cannot meet the requirements for remote applications.

Handheld luminometers designed for hygiene applications are portable, economical and can detect ATP when testing for bacterial contamination. The most sensitive model we found (S1 Table, 3M LX25) is quoted as being sensitive down to 1 femtomole of ATP. Although unit conversion to moles of enzyme depends on the enzyme kinetic rate and other factors, we estimate that this limit-of-detection (LOD) is about 5E-18 moles of luciferase, or about 30-fold above what is required for the split-luciferase assay. Additionally, these devices have incompatible sample formats intended for insertion of long hygienic swabs into the instrument–a practical limitation inhibiting the use of sample formats better suited for serology assays, such as PCR tubes.

Beyond commercial luminometers, the landscape of low-cost, portable luminometers reported in the academic literature cannot match the LOD performance standards established

by PMT-based commercial instruments ([14–22] and S1 Table). In general, we found that the limit of detection of non-PMT devices in the academic literature ranges 100- to 100,000-fold inferior to commercial PMT-based readers. Moreover, many of the academic luminometers did not meet the other required specifications necessary for use in distributed, remote settings.

We designed a low-cost handheld SiPM-based luminometer in order to meet the requirements of remote field applications while achieving a limit of detection that is on par with high-end commercial benchtop instruments. Our device enables assays developed on commercial instruments to be transferred to remote locations, or to be used in distributed applications relevant to global health. The two-channel luminometer is compact and rugged, runs on battery power, and costs less than $700 US in materials. Its design eliminates DC drift in SiPM dark current by using mechanical shutters to modulate and lock in on luminescence signals far smaller than the sensor dark current (Fig 1 and S1 Movie and S2 Movie). Because the demodulated signal is robust to temperature changes, long integration times can be leveraged to achieve improved performance over previously-published SiPM luminometer designs. During development, we encountered a novel, externally-coupled manifestation of the well-known SiPM optical crosstalk effect. We implemented an automated push-button calibration for this additional source of correlated error. Combined with optimized sensor size and strong coupling of the sample to the sensors, our open-source design realizes an approximately 100-fold improved radiometric limit of detection over previously published handheld SiPM-based devices (S1 Table).

We validated the device performance radiometrically, with standardized luciferase enzyme titrations (Fig 2), as well as under varying ambient temperatures (Fig 3). We then deployed the instruments for use in two pilot clinical studies testing for antibody prevalence to SARS-CoV-2 spike (S) and nucleocapsid (N) protein using the split-luciferase assay (spLUC) [8]. In the first study, international serology standard (ISS) antibodies were screened for both (S) and (N) proteins, while fingertip blood was drawn from volunteers in a University of California San Francisco (UCSF) study (Fig 4). Participant samples were screened simultaneously with spLUC, over the counter Lateral Flow Assay (LFA) tests, and Enzyme-Linked Immunosorbent Assay (ELISA), with good correlation observed between all test modalities.

With enabling global health applications as our primary development goal, we conducted a clinical serology study in Mirzapur, Bangladesh, in partnership with the Child Health Research Foundation (CHRF). To support the collaboration and demonstrate scalability, we built a total of ten portable luminometers: eight were shipped to Bangladesh for the study, while two were kept in San Francisco for further testing and validation. A total of 204 volunteers were tested for N- and S- protein sars-cov-2 antibodies in two different field camps, demonstrating performance, ruggedness, and scalability (Fig 5).

## Design of the luminometer

### Choice of sensor type

Low-light detection is typically achieved using photomultiplier tubes (PMTs) and avalanche photodiodes (APDs). While performant, PMTs are not suitable for low cost, rugged, portable devices due to their price, sensitivity to mechanical vibration, and high bias voltage requirements. While APDs are small and mechanically robust, they too are expensive, easily damaged, and also require high bias voltages. Ordinary PIN (p-type—intrinsic—n-type) photodiodes are economical, robust and easy to use, but lack intrinsic sensitivity. Achieving high sensitivity with a PIN photodiode requires excessive external gain, leading to high sensitivity to external electric and magnetic fields. For example, a PIN photodiode-based luminometer device

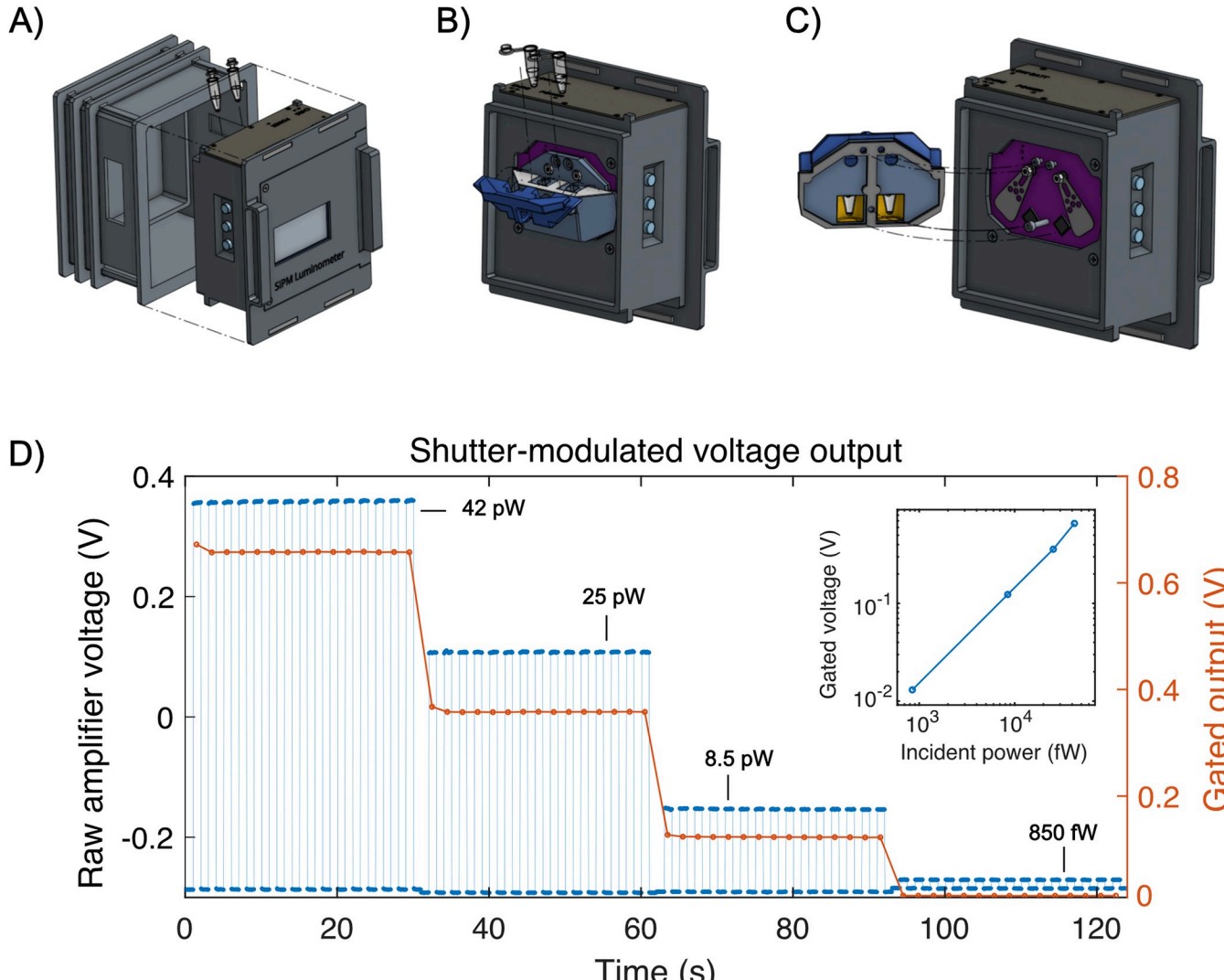

**Fig 1. Design overview and demonstration of operating principle.** A) Overview of the handheld luminometer: A removable rear access lid (left) allows access to the sample holder, accepting two clear, thin-walled PCR tubes. The user interface consists of a display panel, buttons, and a beeper. B) Rear view of the assembly with the access lid removed, showing the tube holder with its hinged lid. PCR tubes containing samples slide downward into the built-in recesses leading to the measurement cavities. C) Rotationally-exploded view of the internal sample cavity and shutter system. The tube holder (left, light and dark blue) houses the sample tubes in their own respective optical cavities coated in reflective mylar film (shown in yellow), increasing optical collection efficiency. The swinging shutter flags are low-profile, laser-cut stainless steel, fitting inside the 2 mm gap between the sensors and sample cavity, in order to preserve the high collection efficiency resulting from positioning the sample as close as possible to the sensor. D) Raw, shutter-modulated voltage traces (blue points, left y-axis) showing signal levels with varying amounts of incident optical power (black text with leaders) produced by filtered light from a feedback-stabilized arc lamp (Excelitas Exacte) imaged onto the luminometer sensor. Individual gated data points (red circles, right y-axis) are computed on the luminometer as described in the main text. Inset: Summary of gated voltage measurements vs. incident optical power, demonstrating the linear relationship between gated signal and incident optical power, which is independent from drift in sensor dark current. Raw data for D) is provided in S1 Data.

described in the literature [17] ultimately had its sensitivity limited by spurious electrostatic charge distributions affecting baseline dark current.

In contrast, SiPMs are compact, economical ($59 at the time of purchase), resistant to damage, exhibit high internal gain, require a moderate bias voltage, and are insensitive to external magnetic and electric fields. SiPM sensors consist of monolithic arrays of microscopic APDs ("microcells"), each with extremely high internal gain, resulting in a large signal amplification for each photon that is detected. Additionally, the arrayed APD format enables a large-area

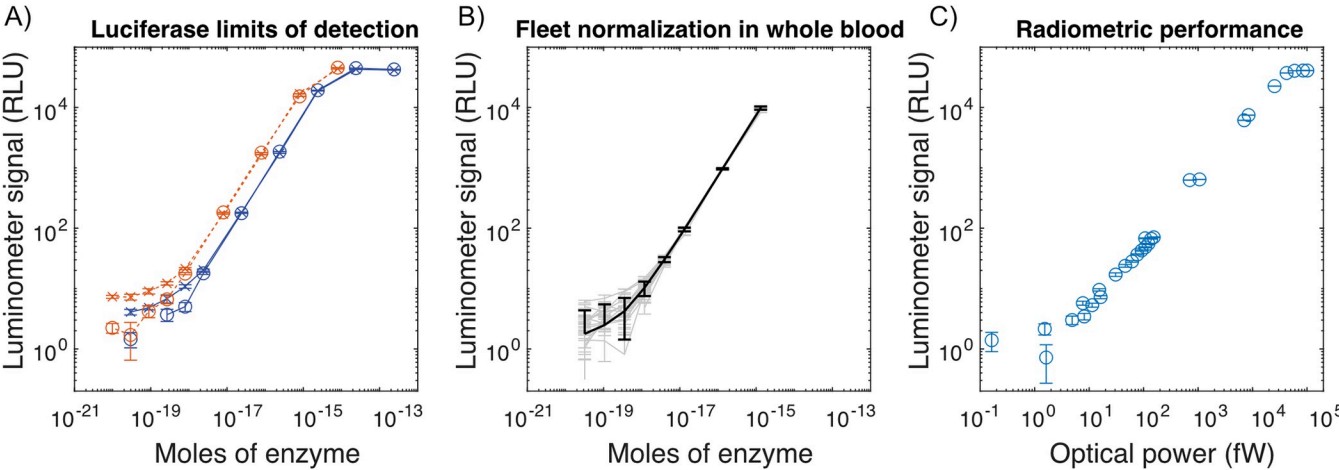

**Fig 2. Basic performance characteristics.** The luminometer's limit of detection is plotted for three contexts. A) Titration of firefly luciferase using luciferase assay system (red dashed lines), and the Bright-Glo assay (solid blue lines). 'X' markers represent channel A and 'O' markers represent channel B; the two channels on the same device. B) To simulate the spLUC assay, the LOD for eight luminometers was validated and normalized by titration of nanoluciferase into whole blood. Individual devices are shown in light gray with internal standard error of the mean (sem) error bars, while the fleet average is shown in black with the standard deviation across devices shown as error bars. Data with random noise fluctuations below zero are not shown due to the log scale. C) Radiometric testing results, shown as luminometer signal vs. incident optical power. In order to visualize the noise floor on a log-scale in the presence of stochastic noise that causes some data to become negative, a small additive offset of 3 RLU was applied to the data in all panels.

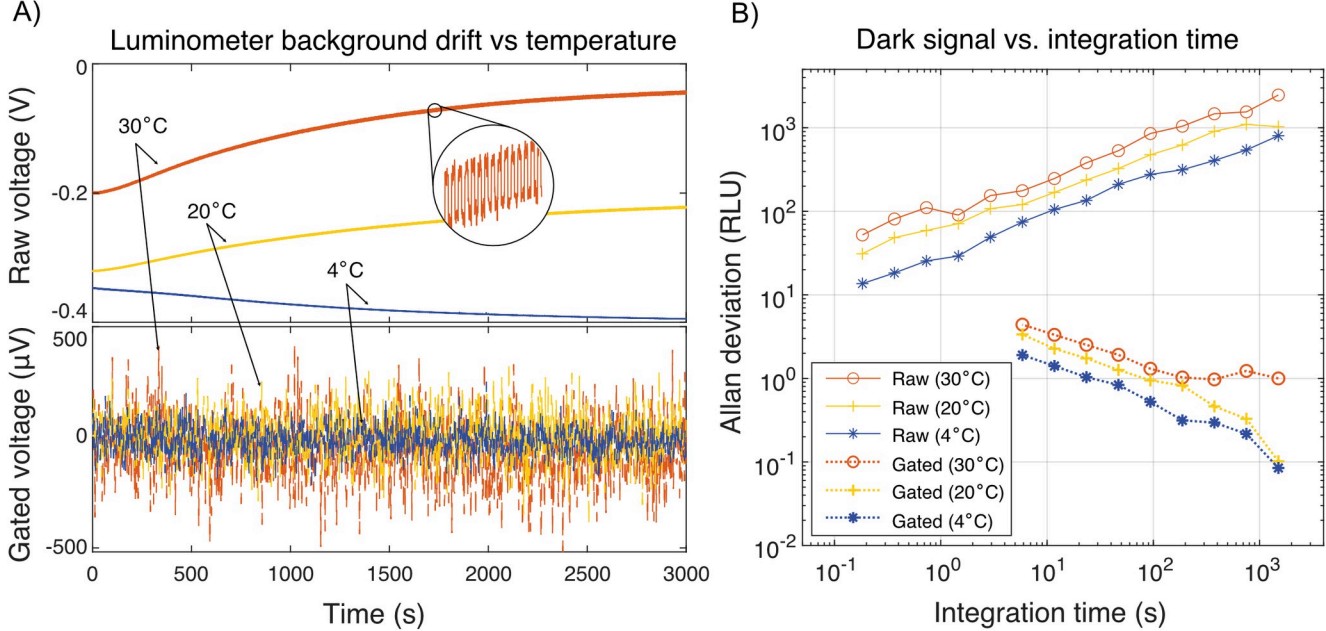

**Fig 3. Performance under temperature stress.** A) Raw output time traces from the transimpedance amplifier are shown as solid traces at three temperatures (top plot), each after a 1 hr pre-incubation of the luminometer at the given temperature. The corresponding gated traces, which include dark current subtraction and ECC compensation, are shown as dotted lines (bottom plot). Gated traces are shown on separate axes due to the difference in amplitude. Inset: Zoom of the raw trace showing the ECC effect in the raw, uncompensated data. B) Allan deviation is used to estimate the lower bound of the instrument noise floor vs. integration time, for all three temperatures. Raw data is shown in solid lines (upper traces) and gated data is shown in dotted lines (lower traces). 1 RLU corresponds to approximately 1 fW of optical power, or ~3E-20 moles of luciferase. Colors in all panels correspond to the ambient temperature.

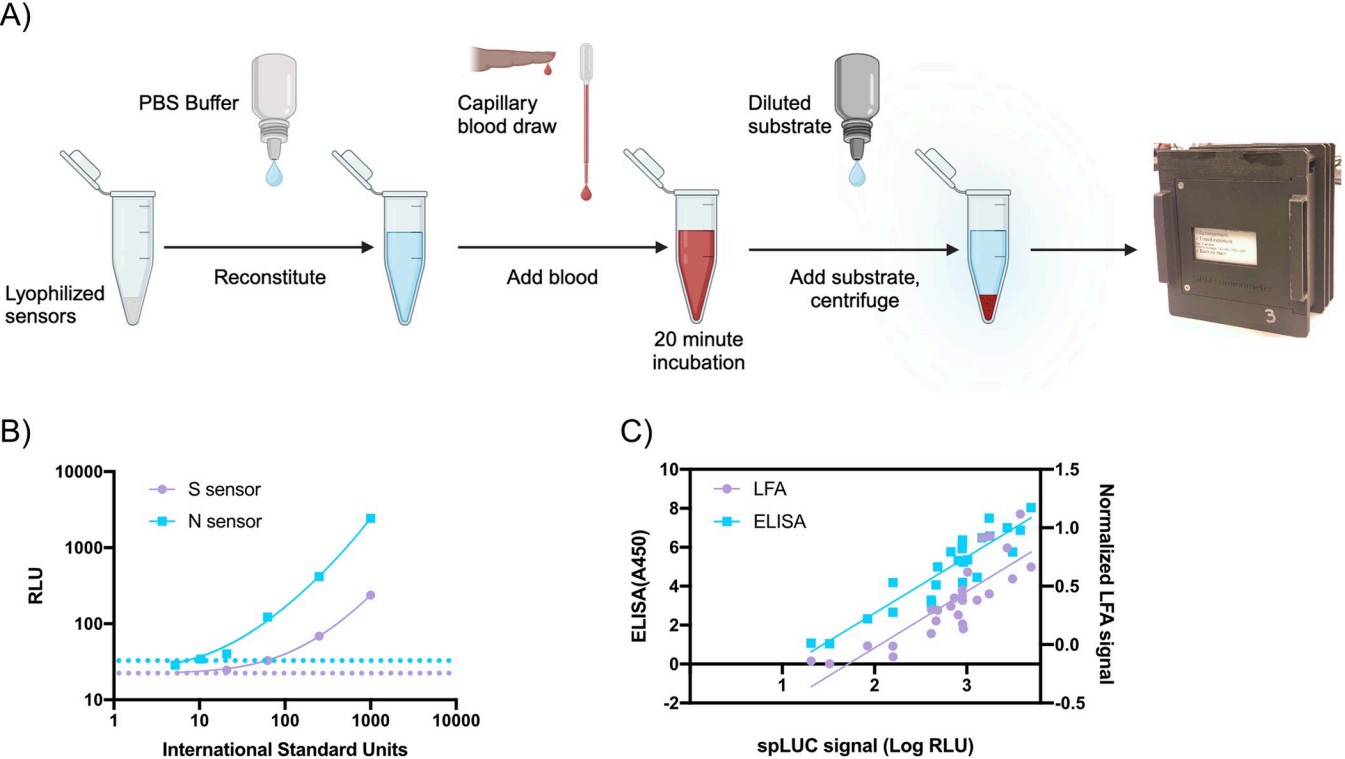

**Fig 4. Detecting antibody levels to COVID-19 with a luciferase-based assay on the handheld luminometer.** A) Schematic of the point-of-care spLUC assay to detect antibodies. Lyophilized protein-based biosensors are reconstituted before the addition of capillary blood, after which luminescent substrate is added, red blood cells are pelleted and the signal is captured on the luminometer. B) Titration of the international serology standard on both the Spike (S) and Nucleocapsid (N) based biosensors. Dotted lines represent the cutoff value for determining a positive signal (purple, S: 22 RLU; blue, N: 32 RLU) as determined by negative control serum. C) Comparison of serology assays on a cohort of 26 vaccinated individuals. The signal from the spLUC assay, which is read out on the handheld luminometer, is compared to the Nirmidas lateral flow assay (LFA) and dried-blood spot based ELISA assay. Both assays show strong correlation to the spLUC assay (LFA: R = 0.84, ELISA: R = 0.91).

sensor format, permitting high collection efficiency from luminescent samples. A well-known disadvantage of SiPM sensors is the magnitude and temperature-dependence of the dark current, which introduces electronic shot noise and low-frequency drift, respectively. In addition to raw dark current, accelerated electrons in SiPMs can spontaneously emit photons that are detected optically in adjacent microcells. This effect is known as "optical crosstalk" [23,24], and further adds to the total dark signal. During the development of this device, we implemented a shutter-based solution to correct for drifting dark current, allowing long integration times to be leveraged without baseline drift. In doing so, we discovered an externally-coupled manifestation of optical crosstalk (ECC), and developed a built-in correction to null the effect. By stabilization over longer integration times and correction of ECC, we have achieved a limit of detection two orders of magnitude lower than previous handheld SiPM-based luminometer designs [15,18,21,22].

Previous SiPM-based handheld luminometers [15,18,21,22] have either made no attempts to mitigate drift in dark current [18,21], or have included peltier-based cooling to stabilize the temperature [15,22]. Although sensor cooling is expected to reduce dark current, the portable device described in [15] reports a radiometric limit of detection (LOD) of approximately 100 fW of optical power– 100× higher than our device, despite their use of a small-area sensor. While the device reported in [22] is compact, portable, temperature-stabilized, and is shown to correlate well with a benchtop luminometer, the report lacks absolute quantification of the

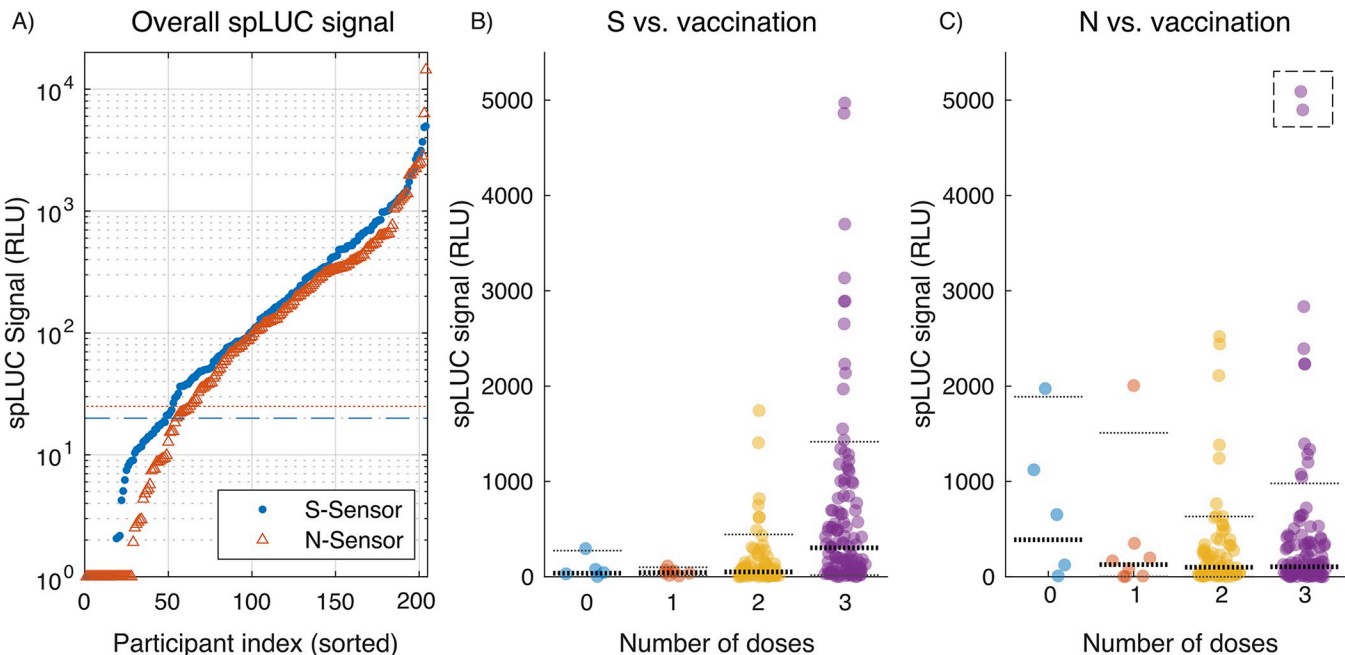

**Fig 5. Bangladesh pilot serology data.** a) Raw S- and N- spLUC signals are individually plotted, sorted in ascending order along with positivity thresholds. Negative values were clipped at a value of '1' for display on the logarithmic axis. The horizontal lines represent the negative cutoff values for each sensor (red dotted line: S-threshold = 20; blue dashed line: N-threshold = 25) that were determined by using mock negative control blood. b) Distributions of spLUC signals for S-sensors are shown as a function of the number of vaccine doses received by each individual. Horizontal lines are used to display quantiles of the distributions, with 10 and 90 percentiles denoted by thin lines, and 50 percentile (median) denoted by bold lines. c) Same as b), but for N-sensors. For clarity, two outlier data points at signal levels of 6,313 and 14,445 RLU were artificially lowered into the displayed range (dashed box).

sensitivity or LOD, and does not include sufficient design detail for reproduction of the results such as design files, bill of materials, or build guide. To our knowledge, this design is the first fully replicable (open-source), handheld, low-cost luminometer capable of a sub-attomole limit of detection for luciferase, allowing demanding assays developed on PMT-based bench-top luminometers to be performed in remote, distributed applications.

## Temporal aspects of signal-to-noise ratio (SNR)

While some previous SiPM luminometer designs emphasized resolving single photons [15,18,20,22], our device was designed with maximal light collection and DC signal integrity as the top priorities. We opted against the added cost and complexity of single photon counting electronics, because it only improves SNR if the knowledge of photon arrival times can be used to distinguish signal events from background events, which is generally not the case, as shown in [18]. Although partial rejection of SiPM optical crosstalk may be achieved by excluding multi-photon events, it was shown previously that SNR was only modestly improved, while both sensitivity and linearity were compromised [18]. Since the dominant source of background events, the sensor dark current, cannot be distinguished from signal on an individual photon basis, achieving a low limit of detection depends uniquely on the long-term DC stability of the system and is not dependent on resolving single photon events at short timescales.

Temporarily disregarding low-frequency thermal drift, the signal to noise ratio (SNR) becomes a simple function of dark current and integration time. The detector's dark current can be characterized by an average overall count rate $\bar{n}_d$ and a stochastic shot noise component

$\sqrt{2\bar{n}_d}$, leading to SNR previously approximated as [25]:

$$SNR = \frac{\bar{n}_{ph}}{\sqrt{\bar{n}_{ph} + 2\bar{n}_d}}\sqrt{\Delta T} \approx \frac{\bar{n}_{ph}}{\sqrt{2\bar{n}_d}}\sqrt{\Delta T} \tag{1}$$

Where $\bar{n}_{ph}$ is the average photon detection rate of the signal component and $\Delta T$ is the integration time. The two terms in the denominator account for the photon shot noise from the signal itself, and from the shot noise of the dark current. The latter contains a factor of 2 due to the uncorrelated subtraction of the dark current with its sampled estimate. Since in the low-light regime the dark current is much greater than the signal itself–in our case $\bar{n}_d$ was approximately between 10–10,000 fold greater than $\bar{n}_{ph}$, the photon shot noise can be safely ignored. With these simplifying assumptions, SNR can be improved by either increasing the total available signal, decreasing the dark current, or by increasing the integration time. The total signal collected is determined by the detector area, quantum efficiency, and collection efficiency, while the dark current depends on the sensor choice and typically scales with surface area.

In reality, low-frequency thermal drift of $\bar{n}_d$ can dwarf all the terms in Eq 1 (Fig 3), nullifying any efforts to improve SNR by signal integration. A successful design must optimize signal integration time, balancing the intersection of thermal drift with reduction of stochastic shot noise in the dark current. In Fig 3, we demonstrate the effect of using a shutter system to eliminate the effects of thermal drift, permitting the limit of detection to become limited mainly by the practical limitations of integration time.

## Choice of sensor size

Eq (1) shows that the signal is approximately linear with the rate of signal photons received, while it is inversely proportional to the square root of the dark current. With dark current proportional to the sensor area, the ideal sensor size is achieved when further size increases result in only a square root increase in photocurrent. Although the exact relationship depends on detailed geometry of the sample and the nearby surfaces, the above condition is approximately true by virtue of the Lambertian emission law [26], when the sensor is marginally bigger than the sample (details in S4 File, where this result is derived). We therefore chose a 6 x 6 mm sensor, accommodating samples volumes up to roughly 165 µL in a 200 µL PCR tube. Further increases in sample volume will contribute to the signal with diminishing returns.

Although cooling may be used to reduce and/or stabilize dark current, we opted against it in this device due to the requirement that it operate in tropical climates with high humidity and dew points, where active cooling of any surface below ambient temperature will result in condensation. Further, active cooling consumes considerable power, reducing battery life, and increases the overall heat dissipation.

## Electronics

The electronics were separated into two distinct boards in order to isolate signal detection from digital noise, motor noise, and heat production. The analog sensor board carries two SiPM sensors, a precision bias voltage circuit, two zero-drift transimpedance amplifiers, and an 8-channel, 24-bit fully-differential ADC readout chip. The digital board interfaces with a Raspberry Pi computer, shutter motor drivers, buttons, and an e-ink display for user interaction. Since the computer and shutter drive components both generate heat and electrical noise, they were kept as far away from the sensors as possible, which were the only components placed on the otherwise flush backside of the analog board facing the sample tubes. The two boards are connected using a 12-pin board-to-board interconnect to transfer power and

digital communication. A digitally-controlled fan was used to remove heat generated internally, with an airflow path that utilizes serpentine inlet and outlet channels to minimize entrance of stray light into the enclosure.

## Mechanical design

Central to our design is a pair of low-profile mechanical shutters used to repeatedly block and unblock light emission from the sample. Signal modulation by the shutters permitted continuous monitoring of the sensor dark current throughout the course of a measurement, allowing longer integration times to be used without impact from temperature-induced drift. Our design functions in a manner conceptually similar to lock-in detection [27], where a signal can be detected in the presence of broadband background noise by virtue of phase- and frequency-locked modulation and detection of the signal. Here, we implement a simple digital filter on the modulated raw data (Fig 1D) that employs the trapezoidal integration method to obtain accurate estimates of the accumulated drift in sensor dark current over time. By subtracting an accurate estimate of the sensor's dark current, the luminescence measurement is free of correlated error and limited by electronic shot noise.

The shutter flags were laser-cut from stainless steel shim stock (OSH Stencil) and mounted between the sensor PCB and small, reflective cavities housing the PCR tubes (Fig 1C). This boosted collection efficiency by redirecting light emitted by the sample in the opposite direction back to the sensor, while the bottoms of the PCR tubes were kept as close as possible to the sensor to maximize direct collection. In order to move the shutters, we designed a mechanism that could reach inside the compact, light-tight sample cavity. We mounted miniature, coreless DC motors in the front compartment of the electronics enclosure, rotating the shutter flags via torque transmitted through machine screws serving as drive shafts. Two sets of miniature ball bearings were mounted between the two PCBs inside a custom machined spacer, holding the screws in place while allowing them to rotate freely about the drive axis. Tight-tolerance, coaxial through-holes in the PCBs provided sufficient clearance for the screws to protrude through both boards, while the close fit of the ball bearings' inner race prevented light from entering the system. The motors were driven by an H-bridge integrated circuit, which was controlled by the Raspberry Pi with a digital PWM signal. The entire device was enclosed in a 3D-printed light-tight enclosure and painted in matte black to reduce stray reflections. Critical surfaces were additionally coated in a commercially-available highly absorbent black paint (Black 3.0, Culture Hustle USA).

## Software and user interface

The user can select from a number of preset signal integration times for a measurement, specified as the number of shutter-open periods (called "samples" in the user interface) to acquire, according to the expected signal level. Each measurement starts and ends with closed shutters. For example, if the total measurement time is set to five samples, then there will be five shutter-open periods and six shutter-closed periods, such that every shutter-open period measurement can be corrected by the average of its flanking shutter-closed periods—equivalent to a trapezoidal integration of the shutter-closed periods over the course of the measurement. During a measurement, the screen displays the cumulative estimated luminescence signal. At the end of the measurement, the screen displays the average luminescence and the standard error of the mean (s.e.m) based on the total number of samples taken.

## Results

We present validation results from both benchtop as well as clinical field study settings. Fig 1 shows the basic structure of the device and the principle of operation. The raw signal is gated

by the shutter in the presence of different optical intensities. Demodulation results in a signal that is directly proportional to the input optical intensity, and unaffected by drift in dark current.

## Performance characterization

**Externally-coupled optical crosstalk.** With the goal of matching the sensitivity of high-end benchtop luminometers, we investigated the performance of the instrument in very low light conditions to gain a deeper understanding of baseline noise and factors limiting sensitivity. With a sample of pure water, we noticed that the sensor current was higher when the shutter was open than when it was closed. Stray ambient light gated by the shutter was ruled out as the cause by recording this effect with the device exposed to ambient room light vs. inside a secondary sealed metal box, which resulted in identical results. Electronic noise emitted from the motors was ruled out by conducting measurements in the absence of shutter flags, which showed no modulation effects on the signal. Finally, shutter-induced stray electric fields were ruled out by grounding and un-grounding the metallic shutter flag to the PCB.

Instead, the modulation was found to be directly correlated with the reflectivity of the shutter flag and that of the sample cavity. Upon further investigation, the residual gated signal was also found to be directly proportional to the raw, temperature-dependent dark current of the sensor. Our findings are consistent with spontaneous optical emission originating from the sensor that is externally reflected back via the sample cavity. We refer to this phenomenon as externally-coupled crosstalk (ECC).

Such spontaneous optical emission is the cause of optical crosstalk, as explained by the sensor datasheet [28] and the primary literature on SiPM. Specifically, charge carriers accelerated by the applied external bias field, called hot carriers, randomly emit photons [23,24,28] which in turn cause secondary electron avalanches within the device. Although the optical crosstalk effect has been discussed exclusively in the context of internal reflections within the silicon and glass cover window [23], we reasoned that the residual shutter-mediated dark signal we observed was likely caused by some fraction of the crosstalk photons escaping the sensor and reflecting back from our system. The magnitude of the effect was found to range from 0.5–0.8% of the raw dark current. Without correction, this effect is responsible for an anomalous offset of up to 75 fW equivalent optical power and a residual temperature drift of about 2 fW/˚C.

We developed a built-in push-button correction to remove the error caused by ECC (S1 Fig). By sampling the effect across a wide variety of temperatures (4˚C to 40˚C), the effect was found to be highly linear with respect to the shutter-closed dark current. The effect could be modeled and subtracted in real time from the raw data. In Fig 3 we demonstrate the robustness of the correction over a range of temperatures.

**Limit of detection.** We benchmarked performance using recombinant firefly luciferase in both Promega Bright-Glo (BG) and Luciferase Assay System (LAS) conditions, representing commonly-used luminescence reference standards comparable between labs and across devices (Fig 2A and S2 Data). We additionally titrated purified nanoluciferase in the presence of whole blood, serving as a means to normalize device performance across our fleet of instruments in the context of a real serology application (Fig 2B and S3 Data). Our limit of detection for firefly luciferase in Promega Bright-Glo assay buffer was 6.4E-19 moles of firefly luciferase, as defined by $LOD = 3\sigma/m$, where $\sigma$ is the standard deviation in RLU across baseline measurements, and m is the measured sensitivity, in units of RLU per mole of enzyme. By the same metric, our limit of detection using LAS was 1.6E-19 moles of firefly luciferase.

We characterized the performance of the luminometer using a home-built radiometric test bench (S2 Fig) in order to validate the performance over a wide range of luminescence

intensities. By using a closed-loop stabilized lamp, radiometric measurements provided data on absolute hardware limitations of the device (Fig 2C and S4 Data). The limit of detection is a function of integration time, which we defined as the number of independent 1s samples: each 1s sample consists of a 1s shutter-open period and a 1s shutter-closed period. With a 150 sample integration, our instrument achieved a limit of detection of approximately 1 fW of radiometric optical power incident on the sensor, or ~2300 photons/s over a 36 mm$^2$ active area. The device is limited in practice by the diminishing return of longer integration times, which become impractical beyond this point, as the SNR improves only with the square root of the number of samples. In Fig 2, we report the limits of detection with both enzyme titration and radiometric testing, and in Fig 3 we report the background noise as a function of temperature, for both raw and gated output.

*Performance under ambient temperature ramp*. Since remote operation in tropical regions is a central application of the device, rejection of temperature-induced drift is central to retaining accuracy. Our temperature compensation scheme consists of several factors: first, the shutter system permits continuous subtraction of the sensor's dark current. Second, our empirical calibration of the ECC effect has proven to be robust to within the system's measurement accuracy for all integration times, and for all ambient temperatures that were explored (S1 Fig). Third, to prevent buildup of excess heat inside the device, a serpentine airflow path was designed into the mechanical layout of the luminometer, such that ambient air flowed over the analog sensor board prior to the main heat-producing elements such as the computer, shutter motors, and the power management board. Despite cooling, we observed shutter-generated changes in the internal temperature of the sensors, although there was no detectable change in the gated signal due to temperature effects.

Further evidence of complete DC temperature drift rejection is presented in Fig 3 (raw data provided in S5 Data), which displays raw amplifier time traces at three different temperatures, after a 1 hr pre-incubation at each temperature. The ECC calibration uses the raw, shutter-closed dark signal to compute the expected measurement offset due to optical crosstalk, which is continuously subtracted from each sampled datapoint. Gated and compensated traces are plotted in Fig 3A, scaled by 100× in order to match the scale of the plot, and exhibit only residual shot noise in the sensor dark current.

In Fig 3B, the Allan deviation of the dark signal reports on the stability of the system across different timescales, by subsampling the data and computing the pairwise deviation of nearest neighbor samples. The raw data exhibits a monotonic increase in deviation with longer integration, meaning that no amount of signal averaging is beneficial without shutter gating. In other words, the raw signal exhibits a random walk. In contrast, the Allan deviation of the shutter-gated and ECC-compensated data monotonically decreases, indicating that the residual error is uncorrelated and converges toward zero with increasing integration time at a slope of -½, consistent with Eq (1). It should be noted that the effect of elevated temperature is not entirely avoided as the amplitude of the thermal dark current increases, and with it the residual stochastic shot noise. The amplitude of the shot noise for a five sample measurement is 1.4 RLU at 4°C, 2.3 RLU at 20°C, and 3.3 RLU at 30°C. Relative shot noise scaling vs. temperature was consistent over the sampled range of integration times.

*Fleet calibration and enzyme titration*. A total of ten luminometers were fabricated and tested. The devices were nominally identical, but required individual testing to verify their performance, as electronic component values can vary within their expected tolerances, and minor differences in assembly may affect the sensitivity. The devices' sensitivities were normalized by preparing a master mix of nanoluciferase diluted to 1 pM (130 μL volume per sample) and measuring it on all devices, each exactly 60 seconds after the final pipetting step when substrate was introduced. Since each luminometer employs a scaling factor used to generate RLU

values, the fleet was normalized to approximately 10 RLU per attomole (1E-18 moles) of nano-luciferase. After normalization, the RLU values measured on all ten devices using samples prepared from the same master mix had a coefficient of variation of 2.45%, demonstrating DC stability and accuracy of the measurements across devices, even in the absence of temperature stabilization.

## Applications

**Using the device for a luminescent antibody detection assay.** With the aim of validating device performance on clinical samples, the luminometer was tested with the spLUC assay for detecting antibodies to SARS-CoV-2, which relies on the fusion of the split nanoluciferase domains to SARS-CoV-2 proteins [8]. The assay is simple and requires no wash steps, making it amenable to non-laboratory or low-resource settings. Additionally, the assay exhibits a large dynamic range and is quantitative across a broad range of antibody concentrations, compared to other serology assays. For example, the ELISA method is a common quantitative lab-based antibody assay but is limited by its multiple wash steps, calibration standard curves, and extensive incubation time (running time 2–4 hours) that makes it impractical in point-of-care applications. The more common lateral flow assay (LFA, running time <30 min) is a very simple nanoparticle method for detecting antibodies but lacks quantitative readout and can produce heterogenous results depending on variations in the manufacturing process [29]. Another assay common in clinical settings is the chemiluminescent microparticle immunoassay (CMIA), however this requires complex antibody functionalized microparticles and specialized instrumentation and that, similar to the ELISA assay, cannot currently be implemented outside of a laboratory. The spLUC assay has the advantage that it is rapid (running time 30 min) and simple, but also delivers a quantitative value for antibody level. However, it requires a highly sensitive luminometer, which is not commonly found in non-laboratory settings. The development of this SiPM luminometer allows for the spLUC assay to be used in rural or low resource settings to obtain accurate and quantitative measurement of antibody levels.

The laboratory-based spLUC assay was previously validated as a point-of-care based assay (described in [8] and illustrated in Fig 4A). To render the sensors stable at room temperature, they were lyophilized in clear, thin-walled PCR tubes. Phosphate-buffered saline (PBS) was dispensed into dropper bottles and estimated that on average 3 drops corresponded to adding 120 µL for reconstituting the sensors. Microsafe capillary pipettes (Safe-Tec) were used to collect 10 µL of fingertip blood and deposit it into the reconstituted sensor tube. At this point, the sample was incubated for 20 minutes to allow the sensors to bind antibodies present in the blood. Ahead of time, nanoluciferase live cell substrate (Promega) was dried by vacuum centrifugation in glass vials to produce a dried substrate. Upon performing the assay, the substrate is reconstituted in water and transferred to a dropper bottle with a transfer pipette. After the incubation of blood and sensors, one drop of reconstituted substrate was added to the sample. While luminescence can be detected in this whole blood suspension, we found that the signal was more than 5-fold higher if the red blood cells were pelleted by centrifugation in order to clarify the sample (S3 Fig). We therefore used a small, battery-powered centrifuge to pellet the red blood cells before detecting luminescence with the SiPM luminometer. Together, this reformatted point-of-care spLUC assay allowed for antibody detection with limited laboratory supplies and simplified steps. We demonstrated that this assay worked by measuring its ability to detect antibodies using the International Serology Standard (ISS) for SARS-CoV-2 developed by the WHO [30]. The point-of-care spLUC assay was able to detect antibodies from the ISS sample over 64-fold dilution for the S sensor and 100-fold dilution for the N sensor, where 1000 International Standard (IS) units represents the undiluted serum standard (Fig 4B). For

serum, the threshold values were determined to be 22 RLU for the S sensor, and 32 RLU for the N sensor by using pre-pandemic negative control serum. The sensitivity with the handheld luminometer was similar to the levels seen on a plate-based benchtop luminometer (see [9], Fig 3).

To demonstrate that the luminometer could be used to detect antibody levels via the point-of-care spLUC assay, we tested capillary blood from 25 vaccinated adult volunteers recruited from the UCSF campus on July 9th, 2021 (S6 Data). We measured antibodies in two replicates via spLUC for both the Spike (S) and Nucleocapsid (N) sensors. The same individuals were tested in parallel via an FDA-approved lateral flow assay for IgG Spike antibodies (Nirmidas MidaSpot), as well as by dried-blood spot Spike ELISA assay [31]. In order to assess correlation between the methods, LFA IgG band intensities were quantified by image analysis and normalized to the intensity of their control bands. Both the LFA and ELISA assay had strong correlation to the spLUC assay signal with R = 0.84 and R = 0.91, respectively (Fig 4C).

**Recording serological prevalence of SARS-Cov-2 in Bangladesh.**   To demonstrate the ability of the device to support global health applications, a pilot SARS-CoV-2 seroprevalence study was implemented using the spLUC assay in the Mirzapur district, 60 km outside Dhaka, Bangladesh, in partnership with the Child Health Research Foundation (CHRF). Eight portable luminometers and spLUC assay reagents were shipped to CHRF and training of personnel to conduct the assay was performed remotely over video chat. A cross-sectional cohort of 204 volunteer participants was assembled from a combination of Mirzapur community members and CHRF staff throughout the period from May-July 2022. Volunteer CHRF staff members were tested from May to through June before rolling the assay out to larger community field camps June 20th and 27th, 2022. Capillary blood samples were collected from each participant by CHRF staff, in addition to vaccination and previous infection history (self-reported, if known). Fingertip capillary blood was used for the spLUC assay to quantify antibodies to both S and N proteins. All raw data for the study is provided in S7 Data.

Our findings indicate that the majority of the 204 participants in the cohort had positive signals for both S and N sensors, with 156 (76.5%) positive for S, 142 (69.6%) positive for N, 185 (90.7%) positive for at least one sensor, and 113 (55.3%) positive for both sensors, defined as signal greater than 20 RLU for S, 25 RLU for N (Fig 5A). These thresholds were determined by using mock negative control blood containing washed red blood cells from a recent donor, reconstituted with pre-pandemic negative control serum. Note that positivity thresholds differ slightly between experiment with serum (Fig 4B) and whole blood (Figs 4C and 5A-5C) due to differences in optical absorption of the sample, and are determined separately (Methods). Although seroprevalence across regional populations may exhibit complex spatiotemporal dynamics, our results are in approximate concordance with previous studies of seroprevalence in Bangladesh residents [32,33].

Although our sample size was small for this pilot study, we were able to detect a robust distribution of antibody titers across the population (Fig 5A). The slope of the numerically-ascending sorted data in Fig 5A reflects the density of points at a given signal level, from which we identify three distinct regions for each sensor: a negative signal region with rapid dropoff (a large relative signal change over a small number of participants), a broad mid-level positive band with signal varying uniformly across a wide range from ~20–1,000 RLU, and a high positive signal band comprising a smaller number of individuals with exceptionally high antibody levels (> 1,000 RLU). Note that due to the nature of the luminometer's strong coupling between its sample cavity and the sensor, the low signal region may be susceptible to small variations caused by differences in blood volume or hemoglobin levels across individuals. In particular, samples lacking SARS-CoV-2-specific antibodies will not exhibit enzymatic emission of light, but will still incur variable absorption of sensor ECC emission by the hemoglobin in

the blood as compared to the calibration sample. Although we mitigate this effect by centrifugation, the pelleted cells nonetheless remain present in the detection volume of the device, but at a reduced cross sectional area. Combined with residual stochastic noise, variation in sampled blood volume and participant hemoglobin density may explain the presence of negative data points in the study, which have been clipped for logarithmic display in Fig 5.

Leveraging participant survey data, we plotted spLUC distributions stratified by the number of vaccine doses received (Fig 5B and 5C). The data reflect a robust response to vaccine boosters for spike antibodies, with the largest change occurring after the third dose. Although the median spLUC value for spike (S) response increases uniformly from one to three doses, the largest observed changes occurred in the 90th percentile of the cohort, which increases from 100 RLU (one dose), to 443 RLU (two doses), and 1417 RLU (three doses). In contrast, the nucleocapsid (N) response is relatively independent of the number doses received, with the median response approximately constant at 127 RLU (one dose), 99 RLU (two doses), and 105 RLU (three doses), and the 90th percentile exhibiting non-monotonic behavior and a high coefficient of variance across all groups. The observed increase in anti-Spike antibodies with vaccine doses may be expected due to the onset of mRNA vaccine administration in the Tangail region in October 2021 [34], which are known to elicit robust responses against spike protein alone. However, earlier vaccination campaigns in the region utilized other vaccines including the Sinovac vaccine which is an inactivated, whole virus formulation that is likely to elicit anti-nucleocapsid antibodies. Indeed, studies of other inactivated SARS-CoV-2 vaccines have been shown to boost anti-nucleocapsid responses in both previously infected and vaccinated individuals, albeit to a lower level than anti-Spike responses [35]. Furthermore, nucleocapsid antibodies have been shown to wane faster than spike-specific antibodies after infection and may explain the lack of correlation between nucleocapsid antibody titers with dose of vaccine [36]. Since there were only five unvaccinated participants in the cohort, there were not enough data to draw conclusions on the effect of a single vaccine dose versus none. Furthermore, the average elapsed time since the most recent dose for those with one and two doses was 150 and 180 days, respectively, whereas those with three received their most recent dose on average 90 days prior to the study date, suggesting titers from the third dose were still high due to the booster's recency.

We recorded self-reported infection status: 28.4% of participants reported having at least one infection, 11.8% having at least two, and 2.9% having three infections. We explore basic statistics stratified by the survey data, in the context of inherent challenges associated with asymptomatic, undetected, or incorrectly-assumed infections, as well as a widely-varying duration and magnitude in individual titers resulting from natural infection [37,38]. For example, a study in the United States [39] found that 43.7% of adults with "serology results indicative of past infection. . .reported never having had COVID-19, possibly representing asymptomatic infection", suggesting that self-reported infection data in general may not correlate well with serological findings. Similarly, previous studies in Bangladesh measured majority seropositivity fractions concordant with our own data [32,33], both of which suggest much higher rates of exposure than indicated by self-reporting.

Nonetheless, the survey data exhibits monotonically-increasing median spLUC values for both S and N sensors, as a function of the number of reported infections (S4 Fig). A Wilcox ranksum test was performed, comparing stratified distributions as a function of dose and infection number, confirming the statistical significance of self-reported infection status, with the greatest significance emerging between zero and any number of reported infections for S signal, as well as the effect of multiple infections on N signal (S5 Fig). In contrast, the N signal was found to be independent of the number of vaccine doses (Fig 5C and S5 Fig), which could be explained by the latter transition to mRNA vaccines with strong anti-Spike responses only.

Our results suggest that the split luciferase assay in combination with handheld luminometers allows for quantitative detection of anti-Spike and anti-Nucleocapsid antibodies in capillary blood, serving as a tool for global health surveillance in pandemic or endemic settings, where acquisition of geospatial health data would otherwise be impractical or too costly to obtain. The pilot study demonstrates the utility of quantitative serological monitoring, permitting downstream analysis and leading to actionable public health information in geographically remote, underserved populations.

## Discussion

We developed a handheld, $700 SiPM-based luminometer that can detect less than an attomole of firefly luciferase–a performance level that is competitive with high-end commercial instruments, and about two orders of magnitude better than previously reported portable, low-cost devices in the literature (S1 Table). Our design makes use of robust, compact, and economical SiPM sensors, enabling the design to be rugged and portable. To achieve a low limit of detection, we employ extended integration times (5–300 s) to overcome residual shot noise in the dark current. Distinct from other SiPM-based designs, we achieve high stability over long exposures by modulation of the luminescence signal using an integrated shutter system. Our open-source design is portable, robust to damage, and battery-powered.

We discovered and overcame an externally-coupled manifestation of the known SiPM optical crosstalk effect, Externally-Coupled Crosstalk (ECC), where optical emission escapes from the sensor, is reflected back and re-detected, causing a temperature-dependent error on the order of 75 fW equivalent power. We built an automated compensation system to null this effect, enabling our technical noise floor to be shot-noise-limited to approximately 1 fW, and in doing so, demonstrated immunity to ambient temperature variation over the range from 4°C—30°C (Fig 3). Although the device is also capable of recording individual measurements in temperatures as high as 40°C, it cannot do so continuously, as the amplifiers begin to saturate from excessive dark current if the shutter motors are stress-tested in continuous operation at 40°C ambient. Our thermal compensation system permits the devices to be operated in changing ambient temperature or humidity, enabling portability through diverse environments that may be encountered on remote, rural health surveys. An equivalent strategy may not be possible with the usage of Peltier cooling elements to stabilize sensor temperature due to condensation in humid environments, in the absence of hermetically-sealed sensor cavities to prevent moisture buildup.

Our design accepts some tradeoffs to attain sub-attomole performance at low cost. Primarily, because we use SiPM sensors with higher dark current than PMTs, longer integration times can limit throughput when processing samples in series, with individual measurements taking 30–60 seconds. Next, the overall mechanical complexity is higher than some previous designs. Our shutter system uses two DC micro-motors, custom gearing, a light-tight PCB mechanical feed-through system, as well as laser-cut steel shutter flags. Assembly and alignment of the shutter system requires moderately complex assembly work. The motors draw power, requiring a substantial battery size (4400 mA-hr), and forced air cooling to prevent buildup of heat inside the box. Our custom electronics require assembly of surface mount components on printed circuit boards, which typically requires a reflow oven. We also made use of industrial quality 3D printing technology (HP Jet Fusion), whose dimensional accuracy and mechanical strength were found to be superior over consumer-grade finite deposition and stereolithography printers. These parts and technologies may not all be accessible to labs or groups around the world wishing to reproduce the design in-house, and such labs may incur extra cost sourcing alternatives. Since we developed the devices for remote deployment with

no internet connectivity, the readout system is entirely manual: the user must record the RLU and error values by viewing them on the display.

In the context of ECC, the absorbance of the sample itself influences the magnitude of ECC coupling and therefore the parameters of the automated calibration. Accurate measurement of samples below the ECC threshold, approximately 75 fW or 2.7 attomoles of luciferase (S1 Fig), must employ a calibration routine using the same volume and optical absorbance of the sample under test. For the split luciferase serology application using whole blood, it was necessary to pre-calibrate the devices using negative control samples (no substrate) in whole blood in order to obtain accurate thermal ECC compensation parameters. This makes it necessary to store user-selectable calibrations on the device that are specific to the sample type and volume.

We recommend that future versions of the device implement improvements related to data handling, power consumption, and mechanical complexity. First, a logical next step would be to develop a smartphone app to accept data wirelessly on a mobile device (by bluetooth for example), which could also record patient metadata, geographical location and other study information, improving efficiency and reducing potential errors and labor of manual data entry. Additionally, the current shutter system uses high pulses of current for fast actuation, drawing significant power and exerting mechanical shock during rapid acceleration/deceleration of the shutter flags (which increases the wear in the mechanism). Instead, a continuously-rotating optical chopper would consume less power, be more robust, and achieve true analog lock-in detection of the luminescence signal, but would require further developments in analog signal processing to implement. With reduced power consumption, the device may not require forced air cooling, making the mechanical design more amenable to light-proofing. Although the current light-proofed design was effective at rejecting indoor or overcast outdoor ambient, it was not able to fully reject direct sunlight entering through the air exhaust port. Finally, improvements to the absolute performance should be possible through the prudent use of sensor cooling, with special attention to the issue of condensation. Our current design contains a forward-looking (unutilized) current-driving module capable of driving two small Peltier elements, which could mount on the rear side of each sensor via PCB clear cuts. Rather than adding temperature sensors for feedback, it should be possible to use the shutter-closed sensor dark current itself as an error signal, which is indeed the quantity of most direct interest. Although sensor cooling will require significantly more development and testing, it holds potential for further improvement in the limit of detection due to a reduction in dark current and ECC magnitude.

We completed two clinical studies with our device, including a pilot-scale serology study in Bangladesh. In doing so, we successfully deployed multiple, sensitivity-normalized instruments to different physical locations and collected robust, quantitative data characterizing the SARS-CoV-2-specific antibody responses of individual participants. Data from two different molecular sensors were acquired, providing information on antibody levels against both spike and nucleocapsid proteins. The resulting quantitative data have potential to either be processed downstream for public health applications, or used on an individual basis to inform on the need for vaccine administration. With regards to portability of the spLUC assay, the cold chain storage of furimazine substrate can be avoided in future work by using a new reagent formulation [40], presenting a way forward for fully room temperature deployment of the spLUC assay.

In summary, we designed a $700 portable luminometer capable of transferring demanding assays from the benchtop to remote field settings. The devices exhibit sub-attomole absolute sensitivities matched across a fleet and are capable of being deployed to rural, difficult-to-access geographic locations simultaneously, with high relevance to global health applications. Additionally, the cost reduction alone as compared to commercial PMT-based devices may

also increase accessibility in centralized locations such as pharmacies, where antibody levels might be recorded during a patient visit to inform on the need for vaccination. For example, in the case of dengue virus, it is particularly relevant to perform pre-vaccination serologic screening [41], suggesting the need for rapid, on-site quantitative patient testing. To increase reproducibility and promote dissemination, we provide a user guide, bill of materials, 3D CAD models, and fully-detailed build guide with instructions.

## Materials and methods

### Ethics declaration

All human samples were obtained under protocols approved by the institutional review boards (IRBs) and in accordance with the Declaration of Helsinki. In both studies, the authors did not have access to participant identifying information.

**UCSF study.**   The University of California San Francisco (UCSF) cohort samples were collected under IRB protocol 20–33062. Samples were de-identified before delivery to the lab where all assays described here were performed. Advertisements including flyers, emails, and word of mouth with information about the study was shared with individuals who were employed at University of California San Francisco (UCSF) and who had received a COVID-19 vaccine. Those interested in the study contacted the study team for more details. All participants were ≥18 years and provided informed written consent.

**CHRF study.**   The study was approved by the ethical review board of the Bangladesh Shishu Hospital and Institute. Participants were enrolled in this study based on their interest in participating and on informed written consent. All participants ≥18 years provided informed written consent. Parental permission and informed written consent were obtained for interested participants aged 16–18 years. Enrolment procedures: Prior announcements were made about the objectives of the study and the need for volunteers. Enthusiastic participants, who communicated with the study staff, and willingly consented to join the study, were contacted for recruitment. Participants who were not CHRF staff received gifts (hand sanitizer and biscuits) for their voluntary participation in the study. Identifying information was not recorded or transmitted to research staff performing assays.

### Luminometer construction

A fully-detailed build guide with instructions, photos, design files, and bill of materials is provided as supplementary information, in addition to the abbreviated summaries below.

**Electronics design.**   PCBs were designed using KiCad and the full design is available here: https://github.com/czbiohub-sf/portable-luminometer/tree/master/Electronics%20design. The PCBs themselves were fabricated by a commercial vendor (OSH Park standard four layer). Surface mount components were soldered to the boards by using a commercial reflow oven (LPKF ProtoFlow S N2). Solder paste was applied using stainless steel stencils (OSH Stencil) and a manual stencil printer (Neodem FP2636). A high temperature solder paste was used on one side of the PCB and low temperature solder paste on the other in order to populate the sides sequentially without component adhesives. Through-hole components were hand-soldered after reflow was completed. SD cards containing the Raspberry Pi OS and custom configurations were cloned from an original disk image. A 4400 mA-hr lithium polymer battery (Adafruit #354) was installed in the enclosure along with a power management board (Adafruit Powerboost 1000C, #2465) that was positioned near the fan-driven airflow exhaust to minimize thermal load. An electronics workstation equipped with a grounded, conductive mat and a stereo microscope was used for all electronics assembly. All analog PCBs were functionally tested prior to soldering of the SiPM sensors. Schematic diagrams of the analog and

digital PCBs are available in S6 and S7 Figs, respectively. Electronic background noise in the absence of a sensor (no dark current) was recorded and found to be substantially lower than any electro-optical effects in the system (S8 Fig).

**Mechanical assembly.** All part numbers and manufacturers are listed in the supplementary information (S1 File). Luminometer enclosures and a majority of the custom mechanical parts were 3D printed using multi-jet fusion technology (Hawkridge technologies, HP MJF 5200 series printer, PA12 material). Individual part files for 3D printing are provided as supplementary material (S5 File). Enclosure parts were painted using a black conductive spray paint (Total Ground paint, MG Chemicals) to reduce stray light and provide electrical shielding. Internal surfaces on potential stray light paths were additionally painted with Black 3.0 (Culture Hustle USA), a highly-absorbent black paint. Tube holders were painted with both Total Ground and Black 3.0, and aluminized mylar film glued into the sample cavities. Shutter flags were custom designed and laser-cut from 0.008" thick stainless steel, and painted with both black paints. PCBs were assembled together using board-to-board interconnects, standoffs, and a custom machined aluminum board spacer containing miniature roller bearings to position and support the shutter driveshafts (M2 machine screws) protruding through both PCBs. The machined spacer additionally served to ground the shutter system electrically, and had machined bosses making contact with ground pads on the digital PCB. Coreless DC drone motors were used to drive shutter actuation (HxChen micro-DC motor). To alleviate space constraints the DC motors were offset laterally and coupled via 3D-printed spur gears with angular cutouts serving as motion stops. Laser-cut rectangular patches of electrical tape were applied to the surfaces of the motion stops to eliminate shutter bouncing at the stops. The drive spur gears were press-fit to the 0.8 mm shaft of the motors and glued with loctite 420. An additional spacer offset the motors axially and served as a motion-limiting stop. The shutter follower gear was threaded onto the drive shaft and secured with loctite 420, and a pair of stainless steel washers coated in teflon dry lube used to reduce friction and improve light shielding against the PCB. Drive shafts were inserted through the assembly, shutter flags attached on the sensor side of the assembly, and clocked rotationally using a custom laser-cut alignment tool. Shutter flags were secured using thin brass M2 hex nuts, tightened with miniature wrenches, and secured using loctite 420. Silicone gaskets were laser-cut and glued into the enclosures to improve light shielding near the button interface, and an open-cell foam gasket was used to pad the display contact area and improve light shielding. The lithium polymer battery was connected to the power management board and secured in place with a custom 3D-printed flex clip.

**Software.** Luminometer control software was written in python and freely available here: https://github.com/czbiohub-sf/portable-luminometer. In brief, a button-operated user menu system was created by connecting callback functions to edge-detection events on the Raspberry Pi GPIO pins, allowing the user to navigate between calibrations, device self-diagnostics, and measurement menus. In the measurement menu, the user is able to select from a range of measurement durations. The software was generally built around a low-latency event loop, which assigned time-consuming I/O-bound hardware interactions to worker threads without delay to the main loop, allowing the button menu to remain responsive throughout long-duration measurements. Data was acquired from the 24-bit ADC via an SPI interface using a custom hardware adapter. To compute each measurement sample, raw data points from the ADC within the same shutter cycle were averaged and gated by subtraction of the mean neighboring shutter-closed points, and further compensated by ECC calibration based on the value of the shutter-closed dark signal. Values from each sample were averaged to produce a final output, with error bars defined by the standard error of the mean (sem) of all the samples. ECC calibrations and other device constants were stored on the device's SD card.

**Radiometric calibration.** A radiometric test station (S2 Fig) was built to characterize the luminometer's optical response in the absence of any variability potentially introduced by changes in enzyme activity levels, liquid settling, the motion of bubbles, or any other effects specific to liquid samples. A closed-loop, aperture-stabilized mercury vapor lamp was used as a light source (Exacte, Excelitas Technologies). The lamp was equipped with a 3 mm Liquid Light Guide (LLG), whose tip served as a stable, homogeneous emission source. The tip of the LLG was imaged onto the luminometer SiPM sensor via a unity magnification 4-f lens relay containing a 2 mm iris at the shared Fourier plane of both lenses, reducing the numerical aperture and intensity of the light. The LLG was additionally filtered using a fluorescence excitation filter (Semrock FF02-475/50-25) in order to restrict the bandwidth of the lamp to a similar emission spectrum as the nanoluciferase enzyme. Finally, a total value of OD = 7.48 in neutral density filters were placed between the second lens and the luminometer to further reduce the light intensity (measured, as opposed to nominal, OD values were used for each filter, by testing with a commercial power meter, Thorlabs P100D). The intensity of the source could then be varied from 0–100% power using the control panel on the mercury lamp. A custom luminometer radiometric rear access lid was 3D printed, into which an optical tube was threaded, providing light-tight coupling of the metered source into the device. The sample holder with reflective cavity was removed from the luminometer to allow the lamp light to reach the sensor. The shutters remained in operation to provide drift compensation. To achieve a range of incident optical powers, the lamp's power setting was adjusted between 0–100% power, and individual neutral density filters were added or removed.

## Luciferase assays

**Preparing Split luciferase sensors, nanoluciferase control, and dried substrate.** The S and N sensors were expressed and purified as previously described in [8] and more detailed in [9]. In brief, all four constructs in the pFUSE vector (S-smbit, S-lgbit, N-smbit, N-lgbit) were transfected into Expi293 cells using Expifectamine according to the manufacturer (Thermo Fisher Scientific). Following purification using Ni-NTA affinity chromatography, the proteins were flash-frozen in liquid nitrogen and stored at -80°C. For lyophilization, each sensor pair (i.e. S: RBD-LgBit + RBD-SmBiT or N protein-LgBiT + N protein-SmBiT) was prepared combined at 48 nM each sensor in PBS + 4.8% BSA + 1.2% Tween pH 7.4. Nanoluciferase enzyme was also expressed via Expi293 cells and purified with Ni-NTA affinity chromatography. Nanoluciferase was prepared at 500 pM in PBS + 4.8% BSA + 1.2% Tween. 5 μL of each mixture was aliquoted into separate PCR tubes, flash frozen in liquid nitrogen, and lyophilized with a batch lyophilizer (SP VirTis BenchTop Pro with Omnitronics, BTP-8ZGEVX, with Crylic Drum 377 Manifold and Bulk Shelf Rack). Live cell nanoluciferase substrate (Promega, N2012) was used to prevent cell lysis during the assay. For the small UCSF cohort study, 57.5 μL of the substrate was vacuum dried (GeneVac) in a black dropper bottle (US Plastic Corps, #66543) and stored at -20°C. In addition, 1192.5 μL of the substrate diluent (from Promega kit) was added to a 3 mL clear dropper bottle. One diluent bottle was prepared per black substrate bottle. For both studies, the lyophilized sensor, control tubes, and diluent bottles were stored at room temperature for at least one month prior to being used. The lyophilized substrate was stored at -20°C until the day of the study.

*Firefly luciferase assays.* In order to compare two different kinetic rates, the enzyme was prepared in both Bright-Glo reagents (Promega #E2610) as well as Luciferase Assay System (LAS, Promega #E1500). Both assays used Promega QuantiLum Recombinant Luciferase (Promega #E1701), and were each performed closely following the manufacturer's respective recommendations. In the former case, the luminescence was stable enough to prepare all conditions and

measure sequentially, with an expected signal decay time of about 30 minutes. In the latter case, special care was taken in order to ensure consistent timing to avoid inconsistent signal decay, which was expected to occur on the timescale of 1–2 minutes. Substrate was only combined, mixed, and immediately measured in a highly consistent manner across all titration points. In both assays, enzyme was serially diluted from stock (as reported previously [8]) into a mixture of phosphate-buffered saline (PBS), 0.2% bovine serum albumin (BSA), and 0.05% tween-20 (PBSTB). A total assay volume of 30 μL was used. For Bright-Glo assays, each measured tube contained equal parts of diluted enzyme in PBSTB (15 μL) and reconstituted substrate in Bright-Glo assay buffer (15 μL), while for LAS, 5 μL of reconstituted enzyme was combined with 25 μL of reconstituted substrate in LAS buffer. All tubes were gently centrifuged for 3–5 seconds prior to measurement to ensure the liquid was at the bottom of the tube and without air bubbles.

*Nanoluciferase titration assay in whole blood.* Nanoluciferase (nanoluc) enzyme was expressed and purified as reported previously [8]. To emulate the spLUC assay performed in whole blood and normalize instrument sensitivity across the fleet of devices, nanoluc enzyme was serially-diluted into a mixture of whole blood and PBSTB (PBSTB-WB, or 12 parts PBSTB, 1 part whole blood) for titration analysis. Whole blood samples were purchased through Vitalant. For each titration point, 280 μL of substrate diluted in Nano-Glo dilution buffer (Promega #N206A) was combined with 1050 μL of diluted enzyme in PBSTB-WB, and divided into a strip of eight PCR tubes, each containing 165 μL of final reaction. Each set of eight tubes (one set per titration point) were measured in parallel across the fleet of portable luminometers.

**Split luciferase assay validation with International Serology Standard (ISS).**   The lyophilized sensors were reconstituted with 3 drops of PBS from a clear 3 mL dropper bottle (US Plastic Corps, #66529, approximately 120 μL). After reconstitution, 10 μL of various dilutions of the ISS (WHO Serology Standard (Mattiuzzo et al. 2020) were incubated for 20 minutes with the reconstituted sensors. Dried nanoluciferase substrate was reconstituted with 1 drop of methanol from a 3 mL clear dropper bottle (US Plastic Corps, #66529). After 5 minutes, all of the previously prepared substrate diluent bottle was added to the reconstituted substrate in the black dropper bottle. Once the sensor and antibody sample incubated for 20 minutes, one drop of the reconstituted luciferase substrate + diluent was added to the sensor sample. The samples were incubated for 5 minutes before measurement on the SiPM luminometer. Negative control serum (Sigma Aldrich, pre 2020) was used to determine the negative cutoff value for serum.

**Split Luciferase assay test on UCSF volunteers.**   The study size was limited by the number of researchers available to process samples, with the number of participants enrolled being sufficient to assess the correlation of spLUC with ELISA and LFA.

For each participant, two lyophilized S sensor tubes and two lyophilized N sensor tubes were tested. In addition, two control nanoluciferase tubes were tested in parallel every hour to account for drift in substrate signal. Two researchers were needed to efficiently run the assay with participants scheduled every 20 minutes. First, one researcher reconstituted the sensors for the upcoming participant (i.e. two S and two N tubes) by adding 3 drops of PBS from the 3 mL dropper bottle to each sensor tube, including the control if applicable (US Plastic Corps, #66529, approximately 120 μL). The tube was flicked down to ensure the liquid reached the bottom and reconstituted the lyophilized pellet. Next, the participants' middle finger was sanitized with an alcohol wipe and allowed to air dry. The finger was pricked with the lancet and kept facing down to let the blood pool. The microsafe capillary pipette (10 μL) gently touched the pooled blood until it reached the filled line. The pipette was then carefully placed into one of the sensor tubes and added by depressing the bulb of the pipette. This was repeated another

three times so that all four tubes contained 10 μL of blood. A new microsafe capillary pipette was used each time. Blood was not added to the control samples. After the addition of capillary blood, all tubes were mixed by rotating up and down. A final flick was done to reduce the amount of blood on the cap of the tube, followed by a 20 minute incubation. For this study, dried substrate was used. During the sensor incubation time, the dried substrate in the black dropper bottle was rehydrated with one drop of methanol (3 mL bottle, approx. 40 μL). After 5 minutes of rehydration, all contents of the substrate diluent (stored in a separate 3 mL bottle) was added to the black dropper bottle with rehydrated substrate. This bottle was rotated up and down to mix. Once the 20 minute sensor incubation completed, one drop of the diluted substrate was added to each sensor tube (and applicable control tubes) and mixed. Sensor samples were then centrifuged with a table-top battery powered centrifuge (TOMY, MPN # 00101244) for 30 seconds to pellet the red blood cells. The samples were then read on the SiPM luminometer with a read time of 30 samples.

**Lateral Flow assay.**    The commercial lateral flow assays were purchased from Nirmidas (MidaSpot COVID-19 Combo Kit). The assays were performed according to the manufacturer's instructions. The package was opened containing a later flow device, capillary tube, sample diluent bottle, and lancet. The middle finger of the patient was cleaned with an alcohol wipe and left to air dry. A lancet was used to prick the finger and the blood was collected with the capillary tube, which automatically stopped at a finite volume. The capillary with blood was then touched onto the white spot labeled "Sample Well" on the lateral flow device until all the blood was absorbed. Then four drops of the sample diluent buffer were added to the sample well membrane. A picture was taken of the result after 20 minutes and the band intensity was analyzed by FIJI image analysis software.

**Dried blood spot anti-Spike antibody ELISA assay.**    Dried blood spots (DBS) were collected from volunteers via capillary blood previously described. The blood was collected into an EDTA tube (BD, #365974). 10 μL was pipetted onto cellulose filter paper (VWR, 05-713-336) and let air dry for at least 1 hour. Afterwards, the DBS were stored at -20˚C. Before the ELISA assay, the dried blood spots were cut into a circle only containing the blood portion, then cut in half, submerged into an eppendorf tube with 100 μL PBS + 0.05% Tween, and left to gently shake at 4˚C overnight (making sure all of the DBS submerged). The next day, the liquid was transferred to a new tube. The ELISA assay was performed as previously described with some modifications [42]. The assay was performed in a 384-well Nunc MaxiSorp flat-bottom plate (Thermo Fisher Scientific), and each sample was run in duplicate. First, plates were coated with 50 μl of 0.5 μg/ml NeutrAvidin mixed in PBS overnight at 4˚C. Plates were then washed three times with PBS containing 0.05% Tween 20 (PBST) and were washed similarly for each of the following steps. Next, 20 μl of biotinylated RBD antigen was added to NeutrAvidin-coated wells and allowed to bind for 30 min at room temperature. After washing, plates were then blocked for 60 min with 80 μl 3% nonfat milk (Lab Scientific)–PBST–10 μM biotin. The reconstituted DBS samples were diluted 1:2 with 1% non-fat milk and 20 μl were added to each well for 1 hour at room temperature. The plates were again washed, and antibodies bound to the coated antigens were detected using 20 μl of protein L -HRP (Thermo Fisher Scientific 32420 [1:5,000]) as indicated for 30 min at room temperature. Following a final wash, the plate was developed for 3 to 10 min at room temperature using 20 μl of 50/50 3,3′,5,5′-tetramethylbenzidine (TMB)/solution B (VWR International). Reactions were quenched with 20 μl 1 M phosphoric acid, and absorbance was measured at 450 nm using a Tecan Infinite M200 Pro spectrophotometer.

*Determining negative cutoff values for point-of-care assay with whole blood.* To determine the cutoff value for S and N sensors, mock negative control blood was constructed. Venous human blood was purchased from Vitalant and 1 mL of blood was washed three times with PBS. The final wash of PBS was removed and pre-pandemic negative control human serum

(Sigma Aldrich, before 2020) was added to bring the volume back to 1 mL. This was done for two blood donors and the sample was run on the point-of-care spLUC assay in combination with the handheld luminometer. The cutoff was drawn above the higher of the two negative control values.

## Split luciferase assay test on Bangladesh participants

The study size was balanced by the availability of CHRF staff and the rate of throughput in the workflow: sample collection, data collection, performing assays, and data entry, while being large enough to demonstrate population health surveillance. The split luciferase assay was performed similarly to the UCSF study described above with minor modifications. Instead of directly collecting blood with microsafe capillary pipettes, capillary blood was collected by scraping into a 0.5 mL EDTA collection tube. The microsafe capillary pipettes were then used to transfer blood from the collection tube into the reconstituted sensor tube. Additionally, the substrate was not dried prior to the study. Instead, the liquid form of the substrate was aliquoted with the same volume (57.5 μL) into a black 3 mL dropper bottle and stored at -20˚C. On the day of the study, the substrate tubes were kept cold with ice packs until the time of use. All samples were run in duplicate, and regular positive control samples were run in order to normalize for possible variation in reagent potency by dividing through by the fold difference in measured control value vs. nominal control value. Final measurements were the average of both control-normalized duplicates.

## Supporting information

**S1 Fig. Sensor externally-coupled crosstalk (ECC) calibration.** Top: Scatter plot of residual gated signal, collected during an ambient temperature ramp from 40˚C to 4˚C, under dark conditions. The residual signal due to the ECC effect is observed to be directly proportional to the raw amplifier voltage, which is in turn linearly proportional to sensor's dark current. The shutter flag modulates the ECC coupling ratio from the sensor back to itself, by blocking and unblocking the reflective sample cavity. The magnitude of the effect represents approximately 150 RLU of correlated error over the recorded temperature range. Bottom: Residuals from the linear fit to the data, showing no evidence of bias across the range of sampled temperatures. (TIF)

**S2 Fig. Cutaway view of the radiometric test station.** A 3D CAD model of the radiometric test setup is shown as a cross-section view. The optical path consisted of a 4f relay, imaging the tip of a 3 mm Liquid Light Guide (LLG) onto the luminometer sensor at 1× magnification. Variable numbers of neutral density filters were added or removed as required to vary the optical power, along with the lamp's percentage output setting (0–100%). The lens relay was constructed using two 100 mm focusing lenses (Thorlabs LA1509-A), and an adjustable iris (fixed at 2 mm) to restrict the NA of the optical path. A 25 nm FWHM spectral bandpass filter (Semrock FF02-475/50-25) was employed to approximately match the wavelength of the lamp's spectrum to that of nanoluciferase emission (460 nm). The incident optical power at the luminometer sensor was varied between 0.1 fW to nearly 1E5 fW. (TIF)

**S3 Fig. Centrifugation of whole blood increases signal in hand-held luminometer.** The spLUC assay was run as previously described and the RLU signal was compared with and without centrifugation of the red cells. Centrifugation increased the signal by 5-fold for this sample from a vaccinated volunteer. (TIF)

**S4 Fig. Bangladesh spLUC results vs. self-reported infection status.** A) S signal vs. self-reported number of infections. B) N signal vs. self-reported number of infections. In both plots, bold dotted lines denote the median of each distribution, and fine dotted lines denote the 10th and 90th percentiles of each distribution. In B), two outlier data points (the same points from Fig 5C in the main text) were artificially lowered into the display range (dashed box), which was limited to 5500 for clarity.
(TIF)

**S5 Fig. Wilcoxon ranksum coefficients comparing spLUC distributions across vaccination and infection status.** Ranksum coefficients denote the P-values of a two-sided Wilcoxon rank-sum test for the median similarity of the sampled distributions, where a value of 1.0 indicates an identical distribution median. Diagonal entries in each matrix compare the same data to itself and all have a value of 1.0. Off-diagonal entries compare distributions from sets of participants with different reported numbers of either doses (A and B) or infections (C and D). Coefficients for the S sensor are shown in A) and C), and for the N sensor in B) and D).
(TIF)

**S6 Fig. Schematic diagram of the analog sensor PCB.** Clockwise, from upper left: the precision bias source uses a zero-drift amplifier and precision low-drift resistors to set the SiPM bias voltage, which controls gain and photon detection efficiency of the sensors. This bias source is powered by an on-board boost converter to generate a 34V supply. The SiPM sensors themselves are read out by fully-differential, zero-drift transimpedance amplifiers, and routed with short traces to a 24-bit differential-input ADC with "global chop" DC bias elimination and digital over-sampling for anti-aliasing. The transimpedance amplifier gain was chosen to position the maximum expected signals near the point of saturation of the amplifier, with parallel capacitors low-pass filtering the bandwidth of the amplified signal. Precision voltage reference sources were used for both the ADC and the differential amplifier. The ADC digitized the signals using an SPI protocol routed directly to a board-to-board interconnect.
(TIF)

**S7 Fig. Schematic diagram of the digital PCB.** Clockwise, from upper left: Connection to the Raspberry Pi zero GPIO header, with labeled pins. H-Bridge circuit for driving the mechanical shutter system. Unused: Buck converter LED driver chips for potential future usage in fluorescence applications, or sensor temperature stabilization (Peltier). Mechanical interconnects: Board to board interconnect (J7) connecting to the analog sensor board, and jumper cables (J1) to the power management board, and the power switch (SW4). Peripherals include: shutter jumper cables (#J1, #J2), user interface buttons (SW1-3), audio indicator (BZ1), fan connector (J5) and its cable (#J3), button caps (#BC1-2) and ground electrodes for the shutter system (TP1-2). Mounting holes were used for PCB mechanical mounting.
(TIF)

**S8 Fig. The baseline noise of the readout electronics is substantially lower than the limit of detection.** Measurements of the transimpedance amplifier noise were acquired prior to installation of the SiPM sensors, in order to characterize the baseline noise in the absence of sensor current. The resulting traces exhibited a total standard deviation on the order of 0.1 RLU at a sampling rate of 20 Hz. With temporal averaging over a period of 30–60 seconds, this noise is further reduced and is more than 100-fold lower than any expected luminescence signals ($> 1$ RLU). The ADC discretization limit was 7.2 nV, or 0.003 RLU.
(TIF)

**S1 Table. Commercial and academic luminometer limits of detection.**
(DOCX)

**S1 Data. Raw data from Fig 1.** Columns include raw data from each luminometer channel for the time series, as well as the associated lamp powers and optical filter values.
(XLSX)

**S2 Data. Firefly luciferase titration data.** Raw data is provided for the firefly luciferase titrations in Fig 2A. Separate tabs denote data from the Bright-Glo assay condition and the Luciferase Assay System (LAS).
(XLSX)

**S3 Data. Fleet normalization with nanoluciferase in whole blood.** Raw data is provided for the fleet titration in whole blood (Fig 2B). The full set of luminometers was used to perform a simultaneous titration of nanoluciferase in order to normalize the sensitivity of all devices. A background of whole blood was used to simulate the spLUC assay condition.
(XLSX)

**S4 Data. Radiometric calibration data from Fig 2C.**
(XLSX)

**S5 Data. Raw data from Fig 3.**
(MAT)

**S6 Data. UCSF Volunteer study data.** Data from Fig 4 is presented in tabular format.
(XLSX)

**S7 Data. Bangladesh serology study data.** All raw data from the Mirzapur serology study (Fig 5) are provided, along with participant self-reported data.
(XLSX)

**S1 File. BOM: Bill of materials.** A full bill of materials is provided for the luminometer. The table is divided into three sections: Assembly items Digital PCB—Itemized Analog PCR—Itemized In the "Assembly items" table, there are columns for 'Internal part number' and also for 'Vendor part number'. When there is only a vendor part number, this is a commercial item that was used as-is in the assembly. When there is an internal part number, the item was fabricated in-house, and the corresponding vendor part number is typically the material used to fabricate the part. Note that we did not attempt an accurate cost estimate for in-house 3D-printed and laser-cut parts. Typically, these parts were very small and had minimal material cost, usually less than $1.
(XLSX)

**S2 File. Guide: Full user guide.**
(PDF)

**S3 File. Guide: Full build guide.**
(PDF)

**S4 File. Note: Discussion of Etendue and optical collection efficiency.**
(DOCX)

**S5 File. 3DP: A.zip archive containing the.stl files for 3D-printed parts in the luminometer.**
(ZIP)

**S1 Movie. Shutter motion across sample.** Computer animation of the shutter motion, shown with all components hidden except for the shutter system, tube holder, and sample tubes. Note that in reality the actuation motion occurs very rapidly (motion is complete after approximately 30 ms), while the stationary phase for open and closed positions is one second in duration. The animation only demonstrates the motion profile.
(MOV)

**S2 Movie. Shutter motion across sensors.** Computer animation of the shutter motion, with only the shutter system and electronics displayed.
(MOV)

## Acknowledgments

The authors are grateful to Georgios Batsios (UCSF) for assistance and usage of equipment for lyophilization of spLUC reagents, and Hawkridge Systems (Sunnyvale, California, USA) for the at-cost production of 3D-printed enclosure parts. We also thank prof. Victor Marcel Acosta (University of New Mexico) for insightful discussions on solid state physics leading to the theory of ECC in SiPM sensors, Greg Courville at CZ Biohub SF for proof-reading the build guide, and prof. Jim Wells (UCSF) for many helpful discussions. The authors are also grateful to the CHRF laboratory team, and the field team for assisting in laboratory assay and sample collection in the field.

## Author Contributions

**Conceptualization:** Paul Lebel, Susanna Elledge, Senjuti Saha, Cristina M. Tato, Rafael Gómez-Sjöberg.

**Data curation:** Paul Lebel, Susanna Elledge, Ilakkiyan Jeyakumar.

**Formal analysis:** Paul Lebel, Susanna Elledge.

**Investigation:** Paul Lebel, Ilakkiyan Jeyakumar, Prasenjit Mondal, Senjuti Saha, Cristina M. Tato, Rafael Gómez-Sjöberg.

**Methodology:** Paul Lebel, Susanna Elledge, Diane M. Wiener, Ilakkiyan Jeyakumar, Emily Huynh, Chris Charlton, Robert Puccinelli, Prasenjit Mondal, Senjuti Saha, Cristina M. Tato, Rafael Gómez-Sjöberg.

**Project administration:** Paul Lebel, Maíra Phelps, Prasenjit Mondal, Senjuti Saha, Cristina M. Tato, Rafael Gómez-Sjöberg.

**Resources:** Paul Lebel, Diane M. Wiener, Cristina M. Tato, Rafael Gómez-Sjöberg.

**Software:** Paul Lebel, Axel Jacobsen.

**Supervision:** Paul Lebel, Senjuti Saha, Cristina M. Tato, Rafael Gómez-Sjöberg.

**Validation:** Paul Lebel, Diane M. Wiener, Prasenjit Mondal.

**Visualization:** Paul Lebel, Susanna Elledge.

**Writing – original draft:** Paul Lebel.

**Writing – review & editing:** Paul Lebel, Susanna Elledge, Diane M. Wiener, Ilakkiyan Jeyakumar, Maíra Phelps, Axel Jacobsen, Emily Huynh, Chris Charlton, Robert Puccinelli, Prasenjit Mondal, Senjuti Saha, Cristina M. Tato, Rafael Gómez-Sjöberg.

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
