## [Decision Letter · Decision Letter 0]

13 Oct 2023

PGPH-D-23-01154

A handheld luminometer with sub-attomole limit of detection for distributed applications in global health

Dear Dr. Lebel,

Thank you for submitting your manuscript to PLOS Global Public Health. After careful consideration, we feel that it has merit but does not fully meet PLOS Global Public Health’s publication criteria as it currently stands. Therefore, we invite you to submit a revised version of the manuscript that addresses the points raised during the review process.

Your manuscript has been evaluated by two reviewers, and their comments are appended below.

Both reviewers are positive towards publication; Reviewer #2 has requested clarification and comment regarding the limit of detection and the practicality of the temperature of storage, as well as the possibility of 3D printing of device components. Please ensure you address each of the reviewer's comments.

We look forward to receiving your revised manuscript.

Kind regards,

Hugh Cowley

Staff Editor

Journal Requirements:

1. We noticed that you used “data not shown" in the manuscript. We do not allow these references, as the PLOS data access policy requires that all data be either published with the manuscript or made available in a publicly accessible database. Please amend the supplementary material to include the referenced data or remove the references.

Additional Editor Comments (if provided):

Reviewers' comments:

Reviewer's Responses to Questions

**Comments to the Author**

1. Does this manuscript meet PLOS Global Public Health’s publication criteria? Is the manuscript technically sound, and do the data support the conclusions? The manuscript must describe methodologically and ethically rigorous research with conclusions that are appropriately drawn based on the data presented.

Reviewer #1: Yes

Reviewer #2: Yes

2. Has the statistical analysis been performed appropriately and rigorously?

Reviewer #1: Yes

Reviewer #2: Yes

3. Have the authors made all data underlying the findings in their manuscript fully available (please refer to the Data Availability Statement at the start of the manuscript PDF file)?

Reviewer #1: Yes

Reviewer #2: Yes

4. Is the manuscript presented in an intelligible fashion and written in standard English?

Reviewer #1: Yes

Reviewer #2: Yes

5. Review Comments to the Author

Reviewer #1: This is a thorough and well written research paper with rigorous validation data in calibration and real sample testing with sound technical contents. In addition, data supported are truly ‘open source’ so that anyone can try to replicate the experiment with relatively low cost.

Reviewer #2: The manuscript details a substantial body of work, both in the design of the apparatus as well as the analysis of it's performance and potential short-comings. The reviewer has the following comments, which would perhaps enhance the manuscript.

1) Limit of detection: Why is the LOD stated for firefly luciferase but not for ATP? A lot of commercial devices in this arena state their LOD for ATP. Whilst it is true that NanoLuc does not need ATP and perhaps there is value in giving information on the LOD with respect to the amount of firefly luciferase, in order to compare devices better, it would be good to get the limit of detection with respect to ATP, at least where firefly luciferase is used.

2) The dried split-luciferase and other sensors are all stored at -20C. I'm unsure how feasible this makes usage in rural areas. Perhaps information on the life-span of these reagents at 4C and/or room temperature would help in decision making/planning beforehand, for those who wish to use this luminometer in a similar fashion.

3) Can any part of the device/manifold be 3D printed? If so, making the programme/script for 3D printing available to the community would be helpful.

Overall, a robust and substantial piece of work.

6. PLOS authors have the option to publish the peer review history of their article (what does this mean?). If published, this will include your full peer review and any attached files.

**Do you want your identity to be public for this peer review?** For information about this choice, including consent withdrawal, please see our Privacy Policy.

Reviewer #1: No

Reviewer #2: No

---

## [Decision Letter · Decision Letter 1]

11 Dec 2023

A handheld luminometer with sub-attomole limit of detection for distributed applications in global health

PGPH-D-23-01154R1

Dear Lebel,

We are pleased to inform you that your manuscript 'A handheld luminometer with sub-attomole limit of detection for distributed applications in global health' has been provisionally accepted for publication in PLOS Global Public Health.

Best regards,

Debjani Paul

Academic Editor

The authors have addressed the points raised by the reviewers in the revised manuscript. Therefore, I recommend acceptance of the manuscript.

Reviewer Comments (if any, and for reference):

Reviewer's Responses to Questions

**Comments to the Author**

1. If the authors have adequately addressed your comments raised in a previous round of review and you feel that this manuscript is now acceptable for publication, you may indicate that here to bypass the “Comments to the Author” section, enter your conflict of interest statement in the “Confidential to Editor” section, and submit your "Accept" recommendation.

Reviewer #3: All comments have been addressed

2. Does this manuscript meet PLOS Global Public Health’s publication criteria? Is the manuscript technically sound, and do the data support the conclusions? The manuscript must describe methodologically and ethically rigorous research with conclusions that are appropriately drawn based on the data presented.

Reviewer #3: Yes

3. Has the statistical analysis been performed appropriately and rigorously?

Reviewer #3: Yes

4. Have the authors made all data underlying the findings in their manuscript fully available (please refer to the Data Availability Statement at the start of the manuscript PDF file)?

Reviewer #3: Yes

5. Is the manuscript presented in an intelligible fashion and written in standard English?

Reviewer #3: Yes

6. Review Comments to the Author

Reviewer #3: The manuscript is of high quality and the authors appear to have addressed the comments of the original reviewers.

7. PLOS authors have the option to publish the peer review history of their article (what does this mean?). If published, this will include your full peer review and any attached files.

**Do you want your identity to be public for this peer review?** For information about this choice, including consent withdrawal, please see our Privacy Policy.

Reviewer #3: No
